# Experiences, needs, and preferences for follow-up after stroke perceived by people with stroke and healthcare professionals: A focus group study

**Emma K. Kjörk**(ORCID)[¤]*, **Carlsson Gunnel, Åsa Lundgren-Nilsson, Katharina S. Sunnerhagen**

Department of Clinical Neuroscience, Institute of Neuroscience and Physiology, Sahlgrenska Academy at University of Gothenburg, Gothenburg, Sweden

¤ Current address: Institute of Neuroscience and Physiology, Department of Clinical Neuroscience, University of Gothenburg, Gothenburg, Sweden

* emma.kjork@neuro.gu.se

**Data Availability Statement:** The data underlying the results of this study are available upon request due to ethical and legal restrictions under the Swedish Secrecy Act 24:8. Interested researchers

## Abstract

### Purpose

The aim of this study was to explore the experiences, needs, and preferences regarding follow-up perceived by people with stroke and healthcare professionals.

### Methods

This is a qualitative exploratory study using focus groups. Patients and healthcare professionals, participating in a clinical visit in primary care or specialised care, were purposively sampled. Data were analysed using a framework of analysis developed by Krueger.

### Results

Focus groups were conducted with two patient groups (n = 10, range 45–78 years) and two multidisciplinary healthcare professional groups (n = 8, range 35–55 years). The overarching theme elucidates *stroke as a long-term condition requiring complex follow-up*. Three organisational themes and six subthemes were identified. People with stroke *discovered feelings and changes after returning home*. In daily life, problems and feelings of abandonment became evident. Participants expressed *experiences of unequal access to health care services*. Barriers for accessing appropriate treatment and support included difficulties in communicating one's needs and lack of coherent follow-up. Follow-up activities were well functioning in certain clinics but did not provide continuity over the long term. Participants made suggestions for *a comprehensive*, *planned*, *and tailored follow-up* to meet patient needs.

### Conclusion

Comprehensive long-term follow-up that is accessible to all patients is essential for equal support. Our findings raised awareness about problems discovered after returning home

may contact the authors at ks.
sunnerhagen@neuro.gu.se or head of department
Jenny Nyström at jenny.nystrom@gu.se for data
access requests. According to Swedish regulation,
interested researchers may only obtain data after
applying and receiving approval from the Swedish
Ethical board.

**Funding:** This study was supported by grants from
the Swedish Stroke Association; the Swedish state
under the agreement of the Swedish government
and the county councils, the ALF agreement
71980; the Local Research and Development
Board for Gothenburg and Södra Bohuslän; the
Swedish Heart and Lung Foundation; the Swedish
Brain Foundation; unconditional grants from
Allergan; Hjalmar Svensson's Foundation; Swedish
NeuroFoundation (Neuro); Rune and Ulla Amlöv's
Foundation; Per-Olof Ahl's Foundation; John and
Brit Wennerström's Foundation; and the Swedish
Science Council VR2017-00946. The funders had
no role in study design, data collection and
analysis, decision to publish, or preparation of the
manuscript.

**Competing interests:** The authors have declared
that no competing interests exist.

and the obstacles individuals face in communicating their needs. Structured follow-up, which is individually tailored, can empower patients.

## Introduction

There is increasing awareness about stroke as a long-term condition, since most people survive a stroke and live many years with residual physical, cognitive, and emotional disabilities [1]. The perceived unfulfilled long-term needs after stroke [2–4] and inequities in access to rehabilitation [5] are challenges that must be addressed. Accordingly, long-term follow-up is essential to identify needs and support long-term recovery as well as the adaptation to life after stroke [6]. Given the complexity of the consequences, the transition process from a specialised to non-specialised environment (stroke-unit at hospital to home and primary care services) creates huge demands on the person with stroke and health care services. Consequences e.g., cognitive impairment [7] could create barriers that interfere with understanding health-related information, communicating one's needs, and taking appropriate actions (often conceptualised as health literacy) [8]. Moreover, current primary care services in Sweden recommend that individuals make their own choices regarding health services and initiate contact if problems occur. These factors lead to a risk of people falling through the cracks in the health care system, resulting in unmet needs and further consequences for the individual [9] as well as for society.

Currently, longer-term support and follow-up after stroke are insufficient in most parts of Europe [10], despite evidence-based recommendations (Global Stroke Guidelines and Action Plan) [6, 11, 12]. A structured follow-up management with a multi-domain approach has been suggested in the Swedish guidelines for stroke; however, this is not broadly implemented in clinical practice. Previous studies have investigated people with stroke and caregivers experiences of health services [13] and the long-term follow-up needs of young and midlife people with stroke [14]. Increased knowledge about people with stroke and healthcare professionals' views regarding needs for follow-up can help optimise stroke services related to a Swedish context. Qualitative research has the potential to improve services by addressing concerns such as practitioner-client interactions and the subjective experiences with disabilities [15]. Accordingly, informing care providers benefits the patients using these services. The aim of this study was to explore the experiences, needs, and preferences regarding follow-up perceived by people with stroke and healthcare professionals.

## Methods

### Study design

This study used an explorative qualitative design with an inductive approach in line with guidelines for qualitative investigation proposing relevance, validity, and reflexivity as overall standards for qualitative inquiry [16]. The underlying philosophical frame of reference in the study relied on social constructivism with a focus on a collective understanding of the participants' views [17]. Focus group discussions were conducted in conjunction with a validation and cross-cultural adaptation process of the Post-Stroke Checklist (PSC) in Sweden [18]. Consolidated criteria for reporting qualitative research (i.e., COREQ guidelines) [19] were followed when reporting the study (see, S1 Checklist). The study was approved by the regional ethical review board in Gothenburg (no. 521–14) and informed written consent was obtained from all participants.

## Participants

Patients and healthcare professionals (nurses, physicians and Occupational Therapist (OT)) were enrolled from primary care or stroke specialised outpatient care at a university hospital. Patients were consecutively recruited and signed an informed consent while visiting a clinical outpatient centre from February 2015 to October 2015. The inclusion criterion was having had a stroke regardless of the time of onset. If cognitive impairment or insufficient knowledge of the Swedish language would make participation in the focus groups unreliable, patients were excluded.

Healthcare professionals working in the targeted outpatient clinics were selected to represent different professions before they were invited and gave informed consent to participate in focus group discussions. Using purposive sampling, we attempted to accomplish homogeneity with respect to having a clinical visit after stroke and heterogeneity [17] concerning time since index stroke, age, gender and profession.

## Data collection and procedure

Data were collected in outpatient clinical facilities and included patient characteristics and focus group discussions. First, during the clinical visit, the health professional invited patients to join in on a focus group discussion. Demographic data (time since index stroke, age, gender) were registered by the healthcare professionals. In addition, patient characteristics (such as type of stroke, neurological characteristics according to the National Institutes of Health Stroke Scale [NIHSS], and activities of daily living dependency) were collected retrospectively from their medical records. Second, the first author (EK) telephoned those available to participate in focus groups and sent them a study information letter including time for appointment. Third, four focus group discussions were carried out, for people with stroke and healthcare professionals separately, to explore their views regarding follow-up.

By using focus group methodology, we sought to stimulate participants to share their views, respond to each other's opinions [20], and possibly raise new topics. Each group met once and the meetings lasted approximately 1.5 hours. The first author was moderator (EK) in all groups and used a semi-structured question guide with open ended questions [20] to lead the discussions. The question guide included the following questions that were complemented by additional questions: What are your experiences with follow-up visits? What problem areas are probably not addressed sufficiently? What do you think is of importance regarding offering good long-term support after stroke? Initially in the meeting, the moderator provided brief information about the study, the focus group approach, and encouraged participants to respond and compare experiences with each other openly to stimulate different opinions. At the end, an oral summary was presented by the moderator to ensure that their contributions were understood correctly. All discussions were audio recorded and transcribed verbatim.

## Data analysis

The focus group data were coded and categorised using the computer software NVivo. Data were analysed by the first and second author following an analysis guide developed by Krueger [20]. The analysis started during the first focus group and was continued thereafter, based on the aim. First, to capture the essential outlines in the discussions, the authors listened to the discussions and read the transcriptions as a whole several times. Second, the data relevant to the aim were identified and grouped into initial categories and subcategories. Data were compared to reveal similarities and differences. Third, a descriptive summary was made of the categories for each group before merging the summaries from all groups. The summaries were descriptive regarding the content of the discussions. Finally, the deeper meaning of the data

was interpreted based on the summaries in combination with transcript quotations. In the analysis process, identified patterns were compared and contrasted across all four groups. The analysis resulted in a thematic structure, (see Table 1 for examples of coding tree). Quotations revealing the ongoing discussions [17] were selected to illustrate the results. In accordance with sampling strategies [21] and when similar discussions repeated, in all groups, [22] data gathering ended.

The first author (PhD student, OT, woman) was the moderator in all focus groups. Multiple coding and discussion of the developing themes were performed in close cooperation with the first and second author (PhD, OT, woman) to highlight alternative interpretations and decisions. Both authors are experienced in qualitative research. All authors have at least 20 years of experience in stroke rehabilitation. The third author (PhD, OT, woman) and last author (PhD, MD, woman) participated by revising and refining the themes.

## Trustworthiness

To ensure the trustworthiness of the qualitative findings, several criteria should be followed: credibility, transferability, dependability, confirmability, and reflexivity [23]. In order to ensure credibility, a range of participants was included in the study representing a variety of consequences after stroke in addition to healthcare professionals with various experiences. Together, this contributed to richness of content in the discussions. Further, in order to enhance the quality of the results, experienced researchers within the stroke field and qualitative research were involved in the analysis. During the analysis, efforts were made to ensure that data were interpreted in context. The transferability of the findings were strengthened by a clear description of the study context, characteristics of the participants, data collection and the steps included in the analysis process. To accomplish dependability the same interview guide was used for all focus groups and discussions were facilitated between the authors during the analysis. Regarding reflexivity, the authors represent different professions and scientific background, giving the possibility for triangulation in the analysis process.

## Results

Forty-six patients and ten healthcare professionals were approached. After purposive sampling, 21 individuals were included, one woman (patient) declined due to a medical condition and two men (health professionals) were not available due to changes in their work positions. Accordingly, 18 individuals participated in the study. Focus group discussions were conducted with people who had a stroke between 3 months and 7 years ago and with a median NIHSS score of 3 at stroke onset. The characteristics of the participants are presented in Table 2.

**Table 1. An example of a coding tree.**

| Quote | Code | Subtheme |
|---|---|---|
| *P7: You're in hospital for a week and then you're just released into the wild. And then there's no specialist or stroke staff there to meet you until you get to see the nurse.*<br>*Moderator: The nurse from the hospital?* | Abandoned | Feelings of emptiness and abandonment |
| *P7: Yes. So it feels like you're just in an empty space.*<br>*P12: Yeah, it's empty there, you're worried, and if* | Empty | |
| *you were worried before, you get even more worried.*<br>*P9: It doesn't get better either.*<br>*P12: Nah, it doesn't get better with time (. . .)*<br>(Group 2. Patients) | Worries | |

**Table 2. Characteristics of all participants in the focus group discussions.** Data are presented as number of persons (n) or median and range.

| People with stroke | Focus group 1 Primary care, rural (n = 4) | Focus group 2 Specialised care, urban (n = 6) |
|---|---|---|
| Age at inclusion | 71 (58–78) | 74 (45–76) |
| Sex, male | 4 | 5 |
| Different country of birth | 0 | 1 |
| Education | | |
| Mandatory | 1 | 4 |
| High School | 1 | 1 |
| University | 2 | 1 |
| Months since stroke | 20 (3–84) | 3 (1–6) |
| Working at stroke onset (yes) | 2 | 1 |
| Length of hospitalization, (days) | 11 (5–82) | 8 (4–11) |
| History of stroke (yes) | 1 | 3 |
| Characteristics at stroke onset | | |
| Ischemic/Hemorrhagic | 4/0 | 4/2 |
| Right/ left/ posterior/ bilateral | 3/1/0/0 | 3/2/1/0 |
| NIHSS | 4 (3–10) | 2 (1–6) |
| Aphasia (yes) | 0 | 1 |
| Neglect (yes) | 1 | 0 |
| At discharge | | |
| ADL independency (yes) | 3 | 6 |
| Wheel-chair use (yes) | 1 | 0 |
| **Healthcare professionals** | Focus group 3, Specialised care, urban (n = 4) | Focus group 4 Primary care, rural (n = 4) |
| Age | 43 (37–46) | 46 (35–55) |
| Sex, male | 0 | 1 |
| Nurse/ OT/ Physician | 3/0/1 | 0/1/3 |
| Stroke experience (years) | | |
| ≤5/ 5-10/ 10 | 0/1/3 | 2/1/1 |

Abbreviations: NIHSS = National Institutes of Health Stroke Scale, ADL = Activities of Daily Living,
OT = Occupational Therapist

A main theme, three organisational themes, and six subthemes were identified in the analysis (Fig 1). The themes were merged based on all groups including both health professionals' and patients' views. The overarching theme underlined that *stroke is a long-term condition that requires complex follow-up*. The organisational themes highlighted that people are *discovering feelings and changes after returning home*, *experiencing unequal access to health care services*, and that participants had *suggestions for a comprehensive, planned, and tailored follow-up*. Selected quotes illustrates the themes and exemplifies discussions also representing other groups.

## Discovering feelings and changes after returning home

**Feelings of emptiness and abandonment.** Patients expressed a sense of emptiness after returning home and a feeling of being left behind (quotation from group 2, see Table 1). Further, due to concerns about having another stroke, their everyday life was perceived to be constrained. Healthcare professionals also expressed similar experiences in their meetings with patients and that a lack of follow-up can lead to increased worries.

**Fig 1. Themes and subthemes derived from the focus group discussions with people with stroke and healthcare professionals regarding the need for follow-up.**

**Becoming aware of changes and problems when back in their everyday life.** When back in their everyday life, patients described changes, both improvements and new problems. Patients and healthcare professionals recognised that several changes do gradually emerge and are often only noticeable in the context of everyday life. In addition, they brought up how hidden problems, such as fatigue or mood changes, can be easily missed. Consequences, such as changed personality and inactivity, were seen as obstacles also affecting their partner's life. Additionally, driving restrictions and changes in work ability were problem areas mentioned. Healthcare professionals emphasised the importance of offering patients an opportunity to talk about problems that were not apparent at discharge.

### Experiences of unequal access to health care services

**Obstacles preventing patients from communicating their own needs.** Both organisational and personal factors were mentioned as obstacles that prevented patients from effectively discussing their needs. Such obstacles could be a lack of knowledge about the condition and treatment options, i.e. regarding mood changes and appropriate training. Further, difficulties knowing *who to* turn to and uncertainty if it was their own responsibility to seek care were barriers for accessing services. Similar discussions arose in health professionals and patients groups exemplified by quotes from Group 1, Patients:

*P14: And I think it's really difficult to work out who is responsible for the whole thing [care after stroke]. I got stuck in the middle.*

*P11: No, you don't know who to turn to for different things.*

*P14: I don't even know which doctor is mine.*

*P11: (. . .) And there's no problem with what each of them does, but I don't think there's any coordination.*

*(agreement)*

*P14: Yes, no, I mean you only get help with the stuff you tell them (. . .).*

Patients having difficulties distinguishing what was due to the stroke and what was associated with normal ageing raised thoughts whether it was appropriate to address certain issues. Other barriers raised by patients were the lack of ability to initiate care, forgetting to mention important things at the follow-up, and difficulty remembering afterwards what had been said. Healthcare professionals also pointed out that cognitive and emotional problems were common in this group, leading to difficulties receiving appropriate patient support.

**No guarantee of coherent follow-up.**   The experiences of follow-up differed within the focus groups as well as between groups. Some patients had good experiences while others felt disappointed. The focus of the follow-up visits was described as different depending on the kind of site, e.g. specialised care or general practitioner, ranging from mainly medical concerns to broad rehabilitation issues. A common perception from the patients was that it was remarkable that a consistent follow-up is not equally accessible to all. Patients who received early supported discharge (ESD) automatically received follow-up. In contrast, patients who did not receive ESD experienced worries and a substantial wait for the ordinary follow-up visit. The differences in follow-up services described by patients were in agreement with healthcare professionals' views.

## Need for comprehensive, planned, and tailored follow-up

**Planned in agreement with the patient before discharge.**   Patients, as well as healthcare professionals, suggested a pre-determined plan for follow-up regardless of stroke severity. Patients agreed that they would like to be called for follow-up, as it may be difficult to make contact themselves. In addition, patients suggested a particular focus on those living alone when planning the long-term follow-up in primary care. Healthcare professionals in specialised care had a clear assumption of the needs and were mainly pleased with their organization for including a follow-up with a nurse at 1 month and a physician at 3 months after discharge. In contrast, physicians in primary care were primarily concerned with the medical issues and were less clear on their understanding of the broader needs. Both patients and Healthcare professionals emphasized the need for follow-up exemplified by quotations from Group 3, healthcare professionals:

*P6: I think it is important that there is a plan for follow-up, so that they know what is going to happen.*

*P15: That they know who they can turn to if something happened or if they got sick again, etc.*

*P5: That they can come and talk about their problems. You don't always know what problems you have until you get home. So I think that it is good with a follow-up and a plan, then they know that these problems are not strange, but they are quite common. Would reduce some worry, etc. (agreement)*

**Individually tailored.**   Healthcare professionals described that some patients had feelings of disappointment regarding follow-up, especially when secondary prevention was the focus.

Patients described a feeling of being lost in their new situation after stroke and raised wishes for advice and professionals asking about their life. Healthcare professionals were aware of the importance of adapting the visit to the patient's life situation and often initiated the visit with a broad question "how are you doing?" However, it was described as challenging to physicians to address patient's individual needs beyond medical conditions and specific areas (e.g. driving, sick leave). In addition, patients often had their own agenda that needed to be addressed within the often short visit. The following are quotations from Group 4, healthcare professionals:

> P10: *We direct our efforts to reduce risk factors, that is the main thing we check with our yearly check-up.*
>
> P3: *Absolutely, and sometimes you find atrial fibrillations, that you maybe don't always do in the hospital. Then there's the question of the driver's licence.*
>
> P1: *Patients sometimes say that it has been very overwhelming and scary and that they sometimes have to wait a long time before they get to see a primary care doctor. (. . .) they are maybe disappointed that they get asked about medicines when they actually have many, larger questions as well.*

## Discussion

Experiences from the focus group discussions yielded a deeper understanding of stroke as a long-term condition that requires complex follow-up. The participants experienced unequal access to follow-up and health care services. Often problems and changes became evident when patients were back in their routines of daily life, leading to feelings of emptiness and abandonment. Participants suggested a comprehensive, planned, and tailored follow-up.

The present findings illuminates concerns patients could face after returning home. The themes 'feelings of emptiness and abandonment' and 'becoming aware of changes and problems when back in their everyday life' have implications for follow-up. Notable, these feelings could be present even though initial symptoms had regressed and a follow-up plan established. Experiences of being abandoned by health services and not being ready to cope in the longer term have been reported previously [2, 13]. This highlights the importance of addressing these emotional concerns [24] and providing the opportunity for all individuals to share their narrative [25] after returning home. The transition from hospital to home can be enhanced if patients are aware of the plan ahead and given ESD [26]. However, in this study, several problems became gradually evident within the daily life context. Therefore, patients might not see the value of support (ESD) or being transferred to primary care in the early phase after stroke. Despite ESD, questions could remain regarding longer-term follow-up and how access to available services can be optimised at later stages.

The theme 'obstacles preventing patients from communicating their own needs' provides information about reasons for not seeking help after stroke. Obstacles could be explained by cognitive and psycho-social issues, as described by both patients and healthcare professionals. This can be discussed within the framework of health literacy. A broad view of health literacy includes having the ability to understand health information, the capacity to take appropriate actions, and expressing one's needs to care providers in order to maintain good health [8]. After a stroke, limited health literacy may affect the ability to access healthcare services. Studies have shown that people feel marginalised and unable to re-engage health care services after a stroke [13]. There can be a range of complex reasons for not seeking help [27]. As a response, the theme 'need for comprehensive, planned, and tailored follow-up' highlights the need for

standardised follow-up to help people with stroke re-access health care services over a longer term. Both individual capacity (i.e. cognition) and environmental support (such as guidelines and services) need to be considered to reduce inequities and empower people with stroke [8]. It is reasonable to believe that a structured follow-up process initiated before discharge could improve access to healthcare and help people with stroke seek help for their own needs.

The problem of unequal follow-up was brought to light in this study within the theme 'no guarantee of coherent follow-up' and exemplified by divergent prerequisites in different settings. Even though rehabilitation is available after the acute phase in Sweden, equal access [5] remains a challenge, especially for problems discovered at a later stage indicating the need for long-term follow-up. Even after a mild stroke or a diagnosis of transient ischemic attack, problems affecting complex activities can be present with a crucial need for follow-up [9, 28].

Healthcare professionals in specialised care were mainly pleased with their current follow-up services in this study. However, the follow-up only comprised the first three months after the stroke and did not provide continuity for longer-term support. Further, the follow-up only included people living in their own homes. In Sweden, as in other countries, there is lack of longer-term services that include holistic and coordinated support [6, 10] beyond the first three months, and this was confirmed in our study. In addition, participants' perceptions regarding new ways of managing long-term follow-up can be expected to be limited in this study. Whether or not people have previous experience with a service may influence preferences and perceived needs [29]. The findings must be understood in this context as people often prefer services they already know; therefore, this should be considered when developing standardised follow-up services.

In the focus group discussions, a comprehensive, planned, and tailored follow-up was suggested for all patients after stroke. However, in current primary care, the focus may be too narrow on issues such as secondary prevention. In addition, addressing one issue at a time is not supported by evidence suggesting a multi-domain approach should be used at follow-up after stroke [1, 6, 30]. Guidelines recommend using tools, such as the PSC [31], to identify long-term needs and facilitate referrals after stroke. Accordingly, a follow-up including the PSC as a standardised structure could meet these needs and support community reintegration and long-term recovery after stroke [12]. In addition, health literacy and the patient's capacity to seek help for their own needs could be enhanced by using a structure such as the PSC in combination with a dialogue with the healthcare professional [18].

## Strengths and limitations

A strength of this study is the sample representing both specialised and primary care and including both patients and healthcare professionals. This contributes to a broad perspective. Even though, healthcare professionals and patients naturally perceive follow-up from different perspectives, the analysis yielded that the deeper meaning of the content was similar. Accordingly, it was possible to merge data and present a thematic structure including all four groups, giving an understanding of the need for follow up after stroke. Participants were selected to elucidate the topic of the study [20]. The sample size in the study was small, but was evaluated as adequate for the purpose of this study in line with accepted recommendations [20]. Larger number of participants or groups that were more diverse could have enriched the content. However, after four focus groups the total richness of the content was evaluated, which was found to satisfy the aim within the study context. To further explore the themes identified in this study and to gain insights of the topic in divergent populations, it could be relevant to conduct larger studies.

Although we attempted to attain heterogeneity in the focus groups, the majority of the patients were male and the majority of healthcare professionals were female. A limiting factor

affecting the sex distribution in the groups was the defined time between enrolment and the focus groups. Nevertheless, heterogeneity was obtained with respect to the patients' stroke characteristics, ages, and education levels and healthcare professionals' ranges of expertise in stroke care and different professions. A limitation is the exclusion of patients evaluated as unable to participate (e.g. cognitive impairment and insufficient knowledge of the Swedish language), accordingly a restriction of the focus group methodology. Further, cognitive functioning was not assessed, which is a limitation since it might have an impact on how they interpreted their experiences and their participation in the discussions. However, the data in present study contributed to a range of opinions to elucidate a complex topic. The homogeneity regarding ethnicity is a limitation of our study since people with different country of birth is representative for the Swedish population as a whole consisting of people with diverse ethnic backgrounds. However, in the discussions, healthcare professionals with extended experience in stroke care contributed to the broader picture, related to the stroke population as a whole.

To enable transferability of the findings, a detailed description of the study context, participants' characteristics, data collection, and analysis were presented in the Methods section. However, the transferability of the findings outside of the Swedish healthcare context could be limited, even though the results contribute to insights that are applicable to other contexts. Further studies are needed to enable a broader representation of the stroke population, i.e. people with different ethnic backgrounds and those who experienced severe strokes who are often overlooked long term. Also, complementing information provided by proxies should be evaluated.

## Conclusions

A comprehensive long-term follow-up service accessible to all patients is essential for equal support. Our findings raised awareness about problems discovered after returning home and individuals facing obstacles in discussing and seeking help for their own needs. Structured follow-up, which is individually tailored, can empower patients.

## Supporting information

**S1 Checklist. Consolidated criteria for reporting qualitative research, the COREQ guidelines were followed when reporting the study.**
(DOCX)

## Acknowledgments

We would like to thank the healthcare professionals and patients for their participation and their willingness to share their experiences. The authors thank Dr. Kate Bramley-Moore for translation help regarding the quotations.

## Author Contributions

**Conceptualization:** Carlsson Gunnel, Åsa Lundgren-Nilsson, Katharina S. Sunnerhagen.

**Data curation:** Emma K. Kjörk, Carlsson Gunnel.

**Formal analysis:** Emma K. Kjörk, Carlsson Gunnel.

**Funding acquisition:** Emma K. Kjörk, Carlsson Gunnel, Åsa Lundgren-Nilsson, Katharina S. Sunnerhagen.

**Investigation:** Emma K. Kjörk.

**Methodology:** Emma K. Kjörk, Carlsson Gunnel, Åsa Lundgren-Nilsson, Katharina S. Sunnerhagen.

**Project administration:** Emma K. Kjörk, Katharina S. Sunnerhagen.

**Resources:** Emma K. Kjörk.

**Supervision:** Carlsson Gunnel, Åsa Lundgren-Nilsson, Katharina S. Sunnerhagen.

**Validation:** Emma K. Kjörk, Carlsson Gunnel, Katharina S. Sunnerhagen.

**Visualization:** Emma K. Kjörk, Carlsson Gunnel, Katharina S. Sunnerhagen.

**Writing – review & editing:** Katharina S. Sunnerhagen.

**Writing – original draft:** Emma K. Kjörk.

**Writing – review & editing:** Carlsson Gunnel, Åsa Lundgren-Nilsson.

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
