## [Decision Letter · Decision Letter 0]

18 Aug 2019

PONE-D-19-16388

Experiences, needs, and preferences for follow-up after stroke perceived by people with stroke and healthcare professionals: a focus group study

PLOS ONE

Dear Mrs Kjörk,

Thank you for submitting your manuscript to PLOS ONE. After careful consideration, we feel that it has merit but does not fully meet PLOS ONE’s publication criteria as it currently stands. Therefore, we invite you to submit a revised version of the manuscript that addresses the points raised during the review process.

ACADEMIC EDITOR: Please insert comments here and delete this placeholder text when finished. Be sure to:

Indicate which changes are required versus recommended for acceptanceAddress any conflicts between the reviewsProvide specific feedback from your evaluation of the manuscript

We would appreciate receiving your revised manuscript by Oct 02 2019 11:59PM. To enhance the reproducibility of your results, we recommend that if applicable you deposit your laboratory protocols in protocols.io, where a protocol can be assigned its own identifier (DOI) such that it can be cited independently in the future. For instructions see: http://journals.plos.org/plosone/s/submission-guidelines#loc-laboratory-protocols

We look forward to receiving your revised manuscript.

Kind regards,

Lars-Peter Kamolz, M.D., Ph.D., M.Sc.

Academic Editor

PLOS ONE

Journal Requirements:

1. Please correct the fist two sentences of the abstract, which seem redundant.

2. Thank you for including your funding statement; "The funders had no role in study design, data collection and analysis, decision to publish, or preparation of the manuscript."

Please provide an amended Funding Statement that declares *all* the funding or sources of support received during this specific study (whether external or internal to your organization) as detailed online in our guide for authors at http://journals.plos.org/plosone/s/submit-now.  

Please state what role the funders took in the study.  If any authors received a salary from any of your funders, please state which authors and which funder. If the funders had no role, please state: "The funders had no role in study design, data collection and analysis, decision to publish, or preparation of the manuscript."

Reviewers' comments:

Reviewer's Responses to Questions

**Comments to the Author**

1. Is the manuscript technically sound, and do the data support the conclusions?

Reviewer #1: Partly

Reviewer #2: Yes

2. Has the statistical analysis been performed appropriately and rigorously? 

Reviewer #1: I Don't Know

Reviewer #2: N/A

3. Have the authors made all data underlying the findings in their manuscript fully available?

Reviewer #1: Yes

Reviewer #2: Yes

4. Is the manuscript presented in an intelligible fashion and written in standard English?

Reviewer #1: Yes

Reviewer #2: Yes

5. Review Comments to the Author

Reviewer #1: Dear authors,

I appappreciate greatly the highly informative study regarding a commonly missed area of interest for the post-stroke care. However, certain modifications should be conducted in order to increase the utilizability of this study:

1- double blinding. The first author participated in both questioning the study participants and in data analysis. There should have been a different analysis team which did not participate in the study to avoid observer bias and other related bias.

2- the cognitive state of the participants and the staff should be elucidated in a more objective way, e.g using Minimental state examination score or any similar semi quantitative cognitive assessment tool as the study is largely related to the cognitive functioning whether in the stroke as a disease or in the mentally-demanding questioning and discussion technique.

3- the multiple and different ethnic groups should be included in the primary study.

Reviewer #2: This is an important field of research. Issues with follow up after stroke are well known and there are now a number of studies which have reported on this. However there is limited data from focus group interviews. Shannon et al (Disabil Rehabil 2016) showed that semi structured interviews add value to questionnaire studies where patients have reported no unmet needs.

This study reports on focus groups using semi-sructured questions and has highlighted a number of important themes that need further input in a Swedish healthcare setting. Many of these can be extrapolated to other developed country health settings.

The main drawback of this study is that the numbers are extremely small and this needs to be highlighted better in the discussion. The findings should be used to inform larger studies to explore the themes identified.

In the abstract, the first section on purpose : please remove the first sentence as both sentences say the same thing.

6. PLOS authors have the option to publish the peer review history of their article (what does this mean?). If published, this will include your full peer review and any attached files.

Reviewer #1: Yes: MM

Reviewer #2: No

---

## [Author Response · Author response to Decision Letter 0]

29 Aug 2019

Comments to the editor:

1. Please correct the fist two sentences of the abstract, which seem redundant.

Thanks, for noticing, it was a mistake. The first sentence is now removed (line 16).

2. Thank you for including your funding statement; "The funders had no role in study design, data collection and analysis, decision to publish, or preparation of the manuscript."

a. Please provide an amended Funding Statement that declares *all* the funding or sources of support received during this specific study (whether external or internal to your organization) as detailed online in our guide for authors at http://journals.plos.org/plosone/s/submit-now. 

Funding statement:

This study was supported by grants from the Swedish Stroke Association; the Swedish state under the agreement of the Swedish government and the county councils, the ALF agreement 71980; the Local Research and Development Board for Gothenburg and Södra Bohuslän; the Swedish Heart and Lung Foundation; the Swedish Brain Foundation; unconditional grants from Allergan; Hjalmar Svensson’s Foundation; Swedish NeuroFoundation (Neuro); Rune and Ulla Amlöv’s Foundation; Per-Olof Ahl’s Foundation; and John and Brit Wennerström’s Foundation. The funders had no role in study design, data collection and analysis, decision to publish, or preparation of the manuscript. 

b. Please state what role the funders took in the study. If any authors received a salary from any of your funders, please state which authors and which funder. If the funders had no role, please state: "The funders had no role in study design, data collection and analysis, decision to publish, or preparation of the manuscript."

The data underlying the results of this study are available upon request due to ethical and legal restrictions under the Swedish Secrecy Act 24:8. Interested researchers may contact the authors at ks.sunnerhagen@neuro.gu.se or head of department Jenny Nyström at jenny.nystrom@gu.se for data access requests. According to Swedish regulation, interested researchers may only obtain data after applying and receiving approval from the Swedish Ethical board.

b) If there are no restrictions, please upload the minimal anonymized data set necessary to replicate your study findings as either Supporting Information files or to a stable, public repository and provide us with the relevant URLs, DOIs, or accession numbers. Please see http://www.bmj.com/content/340/bmj.c181.long for guidelines on how to de-identify and prepare clinical data for publication. For a list of acceptable repositories, please see 

http://journals.plos.org/plosone/s/data-availability#loc-recommended-repositories.

Not applicable

Comments to the reviewers

Reviewer #1: Dear authors,

I appappreciate greatly the highly informative study regarding a commonly missed area of interest for the post-stroke care. However, certain modifications should be conducted in order to increase the utilizability of this study:

1- double blinding. The first author participated in both questioning the study participants and in data analysis. There should have been a different analysis team which did not participate in the study to avoid observer bias and other related bias.

Thanks’ for your comment regarding the analysis of the findings and the potential risk of bias. To avoid bias, this study has followed the COREQ guidelines as required by PLOS ONE and the procedure for focus group methodology described by Krueger. The importance of a systematic procedure and neutrality is emphasized in all steps of the study. To ensure neutrality several actions were applied. The moderator was chosen, in line with guidelines, based on background, training and sensitivity. Further, the researchers’ involvement in both data collection and analyses is part of the procedure and essential for the analysis process. To ascertain all participants views were incorporated in the analysis internal checks were made by the first and second author by reading the transcripts. All points in the results are linked back to the interviews. The first and second author performed the analyses together and finally all authors worked together to ensure neutrality and contextual interpretation. 

In addition, in qualitative research it is important to be transparent about the preunderstanding of the researchers. Therefore, we described previous experiences in stroke and qualitative research. We hope the method section including the subheading trustworthiness elucidates our intention to make appropriate analysis and avoid bias.

We made a change in the following sentence in the method section: 

Line 148: “Multiple coding and discussion of the developing themes were performed in close cooperation with the first and second author (PhD, OT, woman) to highlight alternative interpretations and decisions.

2- the cognitive state of the participants and the staff should be elucidated in a more objective way, e.g using Minimental state examination score or any similar semi quantitative cognitive assessment tool as the study is largely related to the cognitive functioning whether in the stroke as a disease or in the mentally-demanding questioning and discussion technique.

Thanks for your comment. We agree with you that it would have been valuable data, since cognitive functioning might have an impact on participants experiences and discussions. However, in this naturalistic study we did not find it appropriate to extend the inclusion process by adding a cognitive screening. Our attempt was to capture participants’ views and find ranges of opinions to elucidate a complex topic. As long as participants were able to participate in a focus group, we evaluated their opinions as valuable for the discussions. Presence of mild cognitive decline could be present in the sample and is representative for this group of patients.

We have added the following sentence in the limitation section: 

Line 399: “…, accordingly a restriction of the focus group methodology. Further, cognitive functioning was not assessed, which is a limitation since it might have an impact on how they interpreted their experiences and their participation in the discussions. However, the data in present study contributed to a range of opinions to elucidate a complex topic.” 

3- the multiple and different ethnic groups should be included in the primary study.

Thanks for your comment. It could have been valuable to know more about the participants’ background and characteristics. However, in Sweden, data about ethnicity is generally not systematically collected in studies. In this sample, one participant had another country of birth. The homogeneity regarding ethnicity is a limitation of our study since people with different country of birth is representative for the Swedish population as a whole consisting of people with diverse ethnic background.

We added information in the table regarding different country of birth, (page 10)

We have added a sentence in the limitation section as follows: 

Line 403: “The homogeneity regarding ethnicity is a limitation of our study since people with different country of birth is representative for the Swedish population as a whole consisting of people with diverse ethnic background.”

Reviewer #2: This is an important field of research. Issues with follow up after stroke are well known and there are now a number of studies which have reported on this. However there is limited data from focus group interviews. Shannon et al (Disabil Rehabil 2016) showed that semi structured interviews add value to questionnaire studies where patients have reported no unmet needs.

This study reports on focus groups using semi-structured questions and has highlighted a number of important themes that need further input in a Swedish healthcare setting. Many of these can be extrapolated to other developed country health settings.

1. The main drawback of this study is that the numbers are extremely small and this needs to be highlighted better in the discussion. The findings should be used to inform larger studies to explore the themes identified. 

Thanks for highlighting the importance of the study and suggestions for improvement. Most certainly larger studies could contribute to further insights and exploration of the themes in this study. We agree this has to be highlighted in the discussion section and have made changes as follows: 

Line 385: “To further explore the themes identified in this study and to gain insights of the topic in divergent populations, it could be relevant to conduct larger studies”

Line 389: “The sample size in the study was small, but was evaluated as adequate for the purpose of this qualitative study in line with accepted recommendations (20)”.

2. In the abstract, the first section on purpose: please remove the first sentence as both sentences say the same thing. 

Thanks, for noticing, it was a mistake. The first sentence is removed (line 16).

---

## [Decision Letter · Decision Letter 1]

19 Sep 2019

Experiences, needs, and preferences for follow-up after stroke perceived by people with stroke and healthcare professionals: a focus group study

PONE-D-19-16388R1

Dear Dr. Kjörk,

We are pleased to inform you that your manuscript has been judged scientifically suitable for publication and will be formally accepted for publication once it complies with all outstanding technical requirements.

With kind regards,

Lars-Peter Kamolz, M.D., Ph.D., M.Sc.

Academic Editor

PLOS ONE

Additional Editor Comments (optional):

Reviewers' comments:

Reviewer's Responses to Questions

**Comments to the Author**

1. If the authors have adequately addressed your comments raised in a previous round of review and you feel that this manuscript is now acceptable for publication, you may indicate that here to bypass the “Comments to the Author” section, enter your conflict of interest statement in the “Confidential to Editor” section, and submit your "Accept" recommendation.

Reviewer #1: All comments have been addressed

Reviewer #2: All comments have been addressed

2. Is the manuscript technically sound, and do the data support the conclusions?

Reviewer #1: Yes

Reviewer #2: Yes

3. Has the statistical analysis been performed appropriately and rigorously? 

Reviewer #1: Yes

Reviewer #2: N/A

4. Have the authors made all data underlying the findings in their manuscript fully available?

Reviewer #1: (No Response)

Reviewer #2: Yes

5. Is the manuscript presented in an intelligible fashion and written in standard English?

Reviewer #1: (No Response)

Reviewer #2: Yes

6. Review Comments to the Author

Reviewer #1: All comments have been addressed and tackled by the authors. The authors responded in a sound and flexible way

Reviewer #2: (No Response)

7. PLOS authors have the option to publish the peer review history of their article (what does this mean?). If published, this will include your full peer review and any attached files.

Reviewer #1: Yes: MM

Reviewer #2: Yes: Dr Soma Banerjee MD

---

## [Editor Report · Acceptance letter]

23 Sep 2019

PONE-D-19-16388R1 

Experiences, needs, and preferences for follow-up after stroke perceived by people with stroke and healthcare professionals: a focus group study 

Dear Dr. Kjörk:

I am pleased to inform you that your manuscript has been deemed suitable for publication in PLOS ONE. Congratulations! Your manuscript is now with our production department. 

With kind regards,

on behalf of

Dr. Lars-Peter Kamolz 

Academic Editor

PLOS ONE